# MOF-Based Active Packaging Materials for Extending Post-Harvest Shelf-Life of Fruits and Vegetables

**DOI:** 10.3390/ma16093406

**Published:** 2023-04-27

**Authors:** Yabo Fu, Dan Yang, Yiyang Chen, Jiazi Shi, Xinlin Zhang, Yuwei Hao, Zhipeng Zhang, Yunjin Sun, Jingyi Zhang

**Affiliations:** 1Beijing Key Laboratory of Printing & Packaging Materials and Technology, Beijing Institute of Graphic Communication, Beijing 102600, China; 2Beijing Laboratory of Food Quality and Safety, Food Science and Engineering College, Beijing University of Agriculture, Beijing 102206, China

**Keywords:** metal–organic frameworks, active packaging, ethylene absorption/desorption, post-harvest fruit and vegetable, shelf-life

## Abstract

Active packaging that can extend the shelf-life of fresh fruits and vegetables after picking can assure food quality and avoid food waste. Such packaging can prevent the growth of microbial and bacterial pathogens or delay the production of ethylene, which accelerates the ripening of fruits and vegetables after harvesting. Proposed technologies include packaging that enables the degradation of ethylene, modified atmosphere packaging, and bioactive packaging. Packaging that can efficiently adsorb/desorb ethylene, and thus control its concentration, is particularly promising. However, there are still large challenges around toxicity, low selectivity, and consumer acceptability. Metal–organic framework (MOF) materials are porous, have a specific surface area, and have excellent gas adsorption/desorption performance. They can encapsulate and release ethylene and are thus good candidates for use in ethylene-adjusting packaging. This review focuses on MOF-based active-packaging materials and their applications in post-harvest fruit and vegetable packaging. The fabrication and characterization of MOF-based materials and the ethylene adsorption/desorption mechanism of MOF-based packaging and its role in fruit and vegetable preservation are described. The design of MOF-based packaging and its applications are reviewed. Finally, the potential future uses of MOF-based active materials in fresh food packaging are considered.

## 1. Introduction

Fruits and vegetables are necessary for people’s wealth and health, and with the continuous development of means of agricultural production, preservation packaging, and logistics technology, the consumption of fruits and vegetables is increasing worldwide. Consumers are increasingly concerned about the quality of food in terms of nutritional content and freshness, and prefer to purchase food based on its color, aroma, texture, and other characteristics. However, the freshness and appearance of fruits and vegetables are strongly affected by the storage time and preservation conditions [1]. Due to their nature, fruits and vegetables are prone to spoilage and deterioration after picking, leading to economic losses and other problems. About 30% of the world’s food is wasted every year due to spoilage, microbial attack, and mechanical damage [2,3], and this has aroused wide concern around the world. In December 2020, the Food and Agriculture Organization of the United Nations (FAO) called for innovation and technology to promote healthy and sustainable fruit and vegetable production, reduce losses and waste, guarantee product safety and quality, and extend the shelf-life of fresh produce. Therefore, there is an urgent need to develop safe, environmentally friendly food packaging to extend the shelf-life of fruits and vegetables.

Ethylene release is an important factor in the spoilage of fruits and vegetables, and thus it leads to reduced shelf-life. Ethylene is an endogenous plant hormone that has been shown to be associated with the ripening of fruits and vegetables even though it is a gaseous compound, not a phytohormone. Fruits produce large amounts of ethylene after picking, and this promotes fruit ripening, followed by spoilage. The level of ethylene is critical in the freshness of fruits and vegetables, with levels above 0.10 μL^–1^ leading to significant quality losses [4]. The rapid ripening of fruits and vegetables promoted by ethylene is a physiological process that involves changes in chemical, physical and biological processes [5]. The main factors related to ripening include color, hardness, taste, and flavor [6]. Food packaging systems that enable ethylene adsorption/desorption can delay the deterioration effect by decreasing the concentration of ethylene. Conversely, ethylene adsorbed by the packaging can be released to enable ethylene ripening of fruits and vegetables after harvest. Therefore, the development of ethylene adsorption/desorption packaging has received much attention.

Packaging that slows down the release of ethylene from fruits and vegetables has been developed and has been shown to have promising applications. For example, immediate removal of ethylene using oxidizing agents such as potassium permanganate to oxidize ethylene to CO_2_ and H_2_O is an effective way to preserve fruits and vegetables [7]. Shorter et al. [8] adopted alumina-supported potassium permanganate as an oxidizing agent sealed in polyethylene bags to preserve apples; they demonstrated a significant reduction in the level of ethylene in the package, which extended the time of freshness of the apples. Although techniques such as this offer a feasible means of ethylene regulation with potential application in the agricultural industry, the effectiveness of packaging using oxidants to remove ethylene diminishes over time due to oxidant depletion. Other storage conditions such as temperature, humidity, and light (e.g., a dark environment) can slow down the degradation of fruits and vegetables by ethylene to some degree. In addition, and importantly, the chemicals used as oxidants in the packaging are toxic to humans, and this can add a health risk. Therefore, the development of advanced packaging for fruit and vegetable preservation is still a big challenge, and one that affects both supply chain selection and assuring food quality and safety.

Recently, metal–organic frameworks (MOFs) have attracted a great deal of attention due to their excellent gas adsorption and desorption properties, which could be exploited to develop improved packaging for extending the shelf-life of fruits and vegetables [9,10]. Since organic building blocks need to contain groups with accessible free-electron pairs that can bind into metal ions and organize these ions in geometric shapes to produce networks in 2D or 3D rather than discrete molecular units [11]. The excellent controlled release and loading capacity of MOFs has been demonstrated, but few works on its ethylene adsorption/desorption in food packaging applications have been reported [12]. Notably, MOF Technologies teamed up with Decco Worldwide Post-Harvest Holdings to develop the first MOF-based packaging for ethylene adsorption, “Trupick”, in 2016, which prevents fruit and vegetables ripening in storage. Trupick works by releasing 1-methylcyclepropene (1-MCP), a synthetic plant growth regulator that slows down ripening. It was thought that an MOF would be ideal for the storage and release of 1-MCP because of their porous structures.

Compared to the usual methods used to extend the shelf-life of fruits and vegetables, such as the use of ethylene oxidizers and ethylene inhibitors, MOF-based packaging is capable of efficient and selective ethylene adsorption/desorption due to the porous hybrid supramolecular framework of MOFs. This review looks at the use of MOF-based food packaging to extend the shelf-life of fresh fruits and vegetables after harvesting. It focuses on the fabrication, characteristics, and mechanism of ethylene adsorption/desorption bio-based packaging materials. In addition, the challenges to be overcome and future possibilities of the use of MOFs in food packaging are described.

## 2. Fabrication of MOF Packaging Materials

Mesoporous MOF materials are a class of synthetic materials with pore sizes of 2–50 nm, formed by metal ions or ionic clusters bound to organic molecules in a crystal structure that has a large internal surface area (Figure 1). Some MOFs undergo a reversible structural transformation in response to external stimuli, such as object adsorption, temperature, mechanical pressure, light, or an electric field. The structural properties of MOF materials make them promising candidates for gas storage, separation, chemical sensing, and other applications. In 2017, Lashkari et al. [13] investigated the adsorption of allyl isothiocyanate (AITC) molecules by three MOFs, and showed that adsorbed AITC was retained in the pores of the MOFs at low relative humidity (30–35%) and desorption of AITC was achieved at relatively high humidity (95–100%). Recent studies have shown that MUF-16 ([Co(Haip)_2_], H_2_aip = 5-aminoisophthalic acid) is capable of selective adsorption of carbon dioxide from mixtures with other gases such as methane, acetylene, ethylene, ethane, propylene, and propane [14]. This represents a breakthrough for MOF materials and means that efficient and selective adsorption of ethylene gas is also possible in packaging.

### 2.1. Fabrication of Packaging Materials

Researchers have developed a wide variety of synthetic methods, especially in terms of reactors and temperatures. One feature all MOF synthesis methods have in common is crystal growth. An insoluble MOF can be formed from either the solid state, a melt, or a solution. In all cases, an initial nucleation occurs, producing a seed from which smaller or larger crystals may be grown. A key parameter here is the kinetics, which is usually controlled through concentration and temperature [11]. The original method of preparing MOFs was solvothermal synthesis at high temperature and pressure, and a large proportion of MOFs are made in this way. However, mechanochemical, microwave-assisted, sonochemical, electrochemical and diffusion synthesis are also increasingly being used to synthesize MOFs (Figure 2).

#### 2.1.1. Hydrothermal or Solvothermal Synthesis

The organic ligands, metal ions, and reaction solvents are mixed continuously at a specific ratio and temperature. The mixture is then transferred to a high-temperature and high-pressure reactor [15], cooled to room temperature at the end of the reaction, washed continuously with deionized water, and dried under vacuum to obtain the pure MOF material. Zhao et al. [16] prepared Mg-CUK-1, a magnesium-based MOF with high hydrothermal stability, for the adsorption separation of acetylene and ethylene by a hot-solvent method. The MOF materials synthesized in this way have excellent adsorption capacity for ethylene but with no added organic solvents, which significantly reduces environmental pollution during the production process. Zhang’s team [17] used a solvothermal method to synthesize copper terephthalate (CuTPA) MOF-loaded vinyl for encapsulating stored semi-ripe bananas and avocados, and showed that the MOF material significantly accelerated ripening-related color and hardness changes in treated bananas and avocados.

#### 2.1.2. Microwave-Assisted Synthesis

When a solid/liquid mixture is exposed to microwave radiation there is interaction between the mobile charges and the electromagnetic radiation. Microwave synthesis is considered one of the best methods for the synthesis of MOFs due to its cost-effectiveness and advantageous reaction conditions [18]. Kim et al. [19] studied the synthesis of zinc (HBTC) (4,4′-bipy) to 3DMF by solvothermal and microwave synthesis methods, and confirmed that the use of microwave power provided a faster catalyst synthesis while maintaining essential catalytic properties. In addition, the reusability of the catalyst was within the specified range and thus the process is very environmentally friendly.

#### 2.1.3. Sonochemical Synthesis

The matrix solution mixture for the desired MOF structure is introduced into a flared reactor with an ultrasonic bath and no external cooling. The bubble formation and collapse generated around the acoustic waves can produce incredibly high pressures and temperatures in acoustic–chemical synthesis. Yu et al. [20] prepared high-purity and uniformly sized zirconium-based porphyrins MOF-525 and MOF-545 by an sonochemical route. They demonstrated experimentally that the MOF materials showed enhanced hydrolysis of dimethyl-4-nitrophenyl phosphate (DMNP) and more efficient adsorption of bisphenol. It can be used in the detection of food packaging materials in the future.

#### 2.1.4. Electrochemical Synthesis

Electrochemical synthesis combines anode, cell, and cathode plates, with the anode and cathode immersed in an electrochemical medium. Peng et al. [21] have proposed a new electrochemically synthesized Co-MOF/carbon nanohorn-based method for detecting carbendazim (CBZ). A large number of pores and abundant active sites provided a high enrichment capacity for CBZ, which was applied to the detection of CBZ in strawberries and cabbage, with recoveries of 97.25–103.5%.

### 2.2. Forms Incorporated in Food Packaging

MOFs materials can be incorporated in food packaging in three ways: impregnated in the packaging matrix, in sachets, or incorporated in a coating (Figure 3). The incorporation of MOFs in packaging materials has been validated in several studies for gas separation, selective adsorption/desorption, and dynamic treatment of O_2_, moisture, ethylene, or other reactive gases within the package.

#### 2.2.1. Impregnation in the Packaging Substrate

The utility of polymer membranes is always limited by needing to make a choice between permeability and selectivity, while hybrid matrix membranes (HMM) combine the unique properties of selective adsorbance (molecular separation and facilitated gas transport) and the processability and mechanical stability of polymers [22]. Cu-BDC nanosheets [23] were the first materials to be impregnated with a matrix polymer in the form of mixed matrix membranes for the separation of carbon dioxide from CO_2_/CH_4_ mixtures.

#### 2.2.2. Sachets

Ethylene is usually adsorbed by surface molecules, such as potassium permanganate or activated carbon. MOF materials can be used in the packaging of sachets/strips or integrated within the packaging material. Chopra’s team [24] filled polyolefin bags with BasoliteC300 and added the bags to banana packaging. Ethylene-induced banana ripening due to the ethylene bound to the C300 demonstrates that releasing volatile compounds from an MOF is feasible for practical applications.

#### 2.2.3. Incorporation in Coatings

In general, the desired MOF coating materials can be prepared by directly coating them onto suitable substrates, followed by simple heat treatment. Yang’s team [25] chose gelatin hydrogels coated with UiO-66-NO_2_ to prepare robust MOF films after simple heat treatment. The gelatin/UiO-66-NO_2_ film composite hydrogel was found to be more effective in removing lead (II) from apple juice and had little influence on the quality of the apple juice compared with the uncoated film.

**Figure 3 materials-16-03406-f003:**
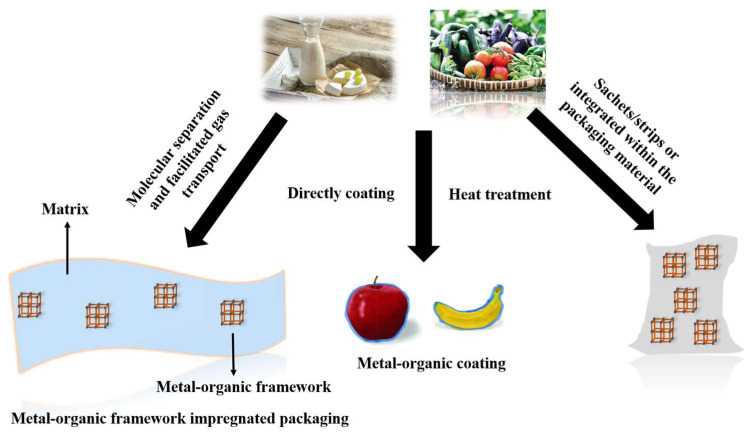
Incorporation of MOFs in packaging material to preserve food quality: MOF-impregnated packaging, sachets containing an MOF, coating with an MOF [26].

## 3. Gas Adsorption of MOF Packaging Materials

The fabrication of innovative packaging is an essential factor in the industrial chain, and adsorption technology for packaging has particular advantages, such as the storage and selective adsorption/desorption of gases. MOF materials are potential gas adsorbents due to their high specific surface area, microporous nature, and pore volume. Hu’s group [27] investigated the gas adsorption of UiO-66(Zr) and found that it can undergo reversible CO_2_ adsorption and desorption more than 500 times, with good resistance to structural damage. Kapelewski’s team [28] fabricated a modified MOF-74 which can store hydrogen through physical adsorption, with a working capacity of 11.0 g∙L^−1^ at 25 °C. Jiao et al. [29] demonstrated selective gas adsorption of HKUST-1, which exhibited better adsorption of xenon (Xe) than krypton (Kr), with an adsorption capacity of more than 60% (mass fraction), which is almost twice that of commercial activated carbon. Therefore, MOF materials as ethylene adsorbents are a crucial development for the packaging industry.

### 3.1. Application of MOF Packaging Materials in Fruit and Vegetable Preservation

In active packaging, an active agent that functions as, for example, a release system, absorption system, or removal system is incorporated in the packaging material to improve the safety and shelf-life of food [30]. Recent studies have highlighted MOFs as potentially active agents in fruit and vegetable preservation because of their excellent gas scavenging [31], antibacterial activity [32], and moisture absorption [33].

#### 3.1.1. Application of MOFs in Ethylene Adsorption/Desorption of Fruit and Vegetable Packaging

Ethylene affects the shelf-life and maturity of fresh agricultural produce and causes spoilage of fruits and vegetables. In food packaging, it is always necessary to avoid ethylene coming into direct contact with the food surface due to safety concerns. Zhang et al. [34] developed MOFs that can be used to encapsulate ethylene and then release it in a controlled manner at the desired stage. It has been proved that ethylene gas can be adsorbed by copper terephthalate MOFs (CuTPA) that have a porosity of 0.39 cm^3^g^−1^ and can adsorb and release up to 654 μL of ethylene [17]. Not only can it create an ethylene-rich space to facilitate the ripening of fruit, but it can also inhibit the spoilage of post-harvest fruits and vegetables and thus extend their shelf-life. Selective adsorption and desorption of chemicals responsible for fruit and vegetable ripening can be achieved with MOFs because of their unique porous structures. In another study, alginate shells combined with MOFs containing aluminum and tricarboxylic acid 1,3,5-benzenetricarboxylic acid ligands demonstrated an uptake capacity of 41 cm^3^g^−1^ for ethylene, while 0.41–0.455 mgL^−1^ of ethylene was released within 3 h [35]. Recently, Zhang et al. [36] evaluated MOF’s potential to be embedded in packaging films as an ethylene adsorber; this study indicates that MgF-embedded LDPE packages could effectively delay banana ripening and extended their shelf life. Significantly, MgF has promising potential as an ethylene adsorber, justifying further work to investigate its application for fresh produce shelf-life extension; this study verified its feasibility under simulated in-transit condition.

However, although MOFs have been proved to be promising candidates for the selective adsorption of ethylene, their dependence on the humidity level for adsorption to occur can, in some cases, limit their application [37]. Awalgonkar’s team [37] found that MOFs have excellent adsorption capacity even under a low relative humidity, which is significantly higher than that of traditional oxidizing agents such as potassium permanganate (KMnO_4_). Moreover, the hydrophilicity of MOFs under conditions of high relative humidity can contribute to the interaction between the MOF and ethylene and increase ethylene adsorption. Chopra’s group [24] investigated the desorption of ethylene and ethylene inhibitor 1-methyl cyclopropane (1-MCP) by Basolite C300 and Basolite A520, and showed that Basolite C300 had a good ripening effect on agricultural products when the package had a high water content.

#### 3.1.2. Combination of Different Materials in MOFs

MOFs for food packaging applications can be combined with other materials that are used to extend the shelf-life of post-harvest fruit and vegetable products in different ways (Table 1). Guan et al. [35] developed an alginate shell containing MOF cores as a packaging system. The MOF cores were charged with ethylene and encapsulated in compact beads formed in the alginate-Fe(III) matrix. Degradation of the alginate-Fe (III) matrix when it was exposed to aqueous sodium citrate solution triggered the release of ethylene. Zhang’s team [17] developed a solid porous MOF to encapsulate ethylene gas for subsequent release. Synthesized copper terephthalate (CuTPA) MOF was loaded with ethylene in sealed containers containing bananas and avocados. The results showed that the CuTPA MOF, which had a highly porous structure, released up to 654 μL/L of ethylene for use with post-harvest agricultural products. In another study, Li et al. [38] developed a CD-based MOF material (α-CD-MOF-Na and α-CD-MOF-K), which not only enhanced the adsorption capacity of ethylene but also the storage stability. The encapsulation capacity of the synthetic material was much higher than that of a single material. In addition to the adsorption of ethylene gas, MOFs can also adsorb other organic volatile gases. Kathuria et al. [39] prepared an MOF material containing bio-based CD and alkali metal ionic groups, which can be used to encapsulate ethanol as a non-toxic adsorbent material. The highest content of CD-MOF adsorbed ethanol was 20 g per 100 g.

### 3.2. Selective Adsorption/Desorption Mechanism of Ethylene in Packaging

Typically, the environment of agricultural products in packaging materials consists of multiple gases such as ethylene, water, and others. Importantly, ethylene biosynthesis affects the growth cycle and respiration rate of plants, which can alter their physical and chemical stability [40]. Because of the complicated gas environment, the selective adsorption and capacity of ethylene is vital for the packaging materials to be effective. Different types of fruits and vegetables have different sensitivities to ethylene and, therefore, ethylene has a different effect on their bioreactions (Table 2). Fresh fruits can be divided into leapfrog and non-leapfrog types according on their ripening mechanism and ripening behavior [6]. Leapfrog fruits and vegetables such as apples, peaches, and avocados usually have a high rate of ethylene production and are highly sensitive to ethylene gas, whereas non-leapfrog vegetables (broccoli, cauliflower) and fruits (cherries, grapes) maintain ethylene emission concentrations at basal concentrations because their respiration rate does not change significantly [6,41]. All changes in leapfrog fruits, such as color, hardness, taste, and flavor, are regulated by ethylene, which acts continuously in the metabolic process because fruits are active organisms, and product deterioration can be influenced by intrinsic characteristics of the fruit and the storage environment. Most importantly, the sensitivity of plant tissues to ethylene is inextricably linked to the time of contact with the atmosphere and temperature (Table 3). Therefore, the selectivity of ethylene of MOF packaging materials is a critical factor.

#### Discussion of Selective Adsorption/Desorption Mechanism

MOFs can selectivity adsorb different gases depending on the pore size and chemical properties of the MOF. A gas can be selectively adsorbed by a metal–organic framework in three ways: adsorbent–adsorbent interaction, molecular sieve effect, and stimulated response gate opening [42]:Adsorbent–absorbent interactions. The sorbent–adsorbent interaction is an affinity between the inner surface and the sorbent in the MOF. This affinity may be due to the interacting van der Waals forces. As a MOF has different adsorption sites, the distribution of the charge and electron cloud can change. Dutta et al. [43] showed computationally that chemical bonds form through the van der Waals force interaction of the aldehyde “tail” with the MOF junction. The different geometries of the metal sites and the pores of the MOFs they studied provided different contributions of the bonds to the adsorption energetics.Molecular sieve effect. Since MOFs are composed of nano-scale pores, there is a possibility of a molecular sieve effect, which means that MOFs can selectively adsorb specific gas molecules. The performance depends on the pore size or channel size. A study has shown that, due to their pore structure, MOFs can selectively adsorb nitrogen from mixtures containing ethylene [44].Stimulus Response Gate. When a stimulus such as temperature [45], pressure [46,47], or light triggers the MOF response gate, the gate is opened and gas molecules enter the pores of the MOF [48,49]. In a study of the photosensitive properties of an MOF, the diaryl ethylene-azobenzene metal–organic backbone showed different adsorption properties at different sites [50].

Although the mechanism of the scavenging of ethylene by MOFs is not well understood, many researchers suggest that it is mainly due to electrostatic interactions between positively charged metal ions and the pelectrons of the ethylene molecules [37]. Li’s team reported the use of Fe_2_(O_2_)(dobdc), an MOF material containing Fe-peroxy sites, to separate ethane/ethylene mixtures. They found that the Fe-peroxy sites in this MOF have a strong interaction with ethane, leading directly to a polymer grade 99.99% pure ethylene product from the ethane/ethylene mixture. In the absence of Fe-peroxy sites, the Fe_2_(dobdc) MOF becomes biased to adsorb more ethylene by opening the iron sites [51]. Lin et al. reported the synthesis via calcium nitrate and square acids of ultramicroporous MOFs (Ca(C_4_O_4_)(H_2_O) (also known as UTSA-280) with rigid one-dimensional channels. The size of the ultra micropores is similar to ethylene molecules, and so these MOFs can act as molecular sieves to allow the passage of ethylene while preventing the passage of ethane molecules. To evaluate the feasibility of using UTSA-280 for gas separation, Li et al. performed gas penetration experiments using a quaternary methane/ethylene/ethane/propane mixture. They found that there was specific enrichment of ethylene in the quaternary gas mixture due to the sieving effect. They also demonstrated the ability of this MOF material to selectively enrich ethylene in complex cracking streams [52]. This research progress has paved the way for advancing the use of MOFs with selective adsorption in agricultural food packaging. Although ethylene is known as the fruit-ripening hormone, other chemicals such as acetylene are also effective fruit ripening molecules. Yang et al. [53] obtained materials to boost the molecular sieving-based separation of CO_2_/C_2_H_2_ and realized the overwhelming adsorption of CO_2_ over C_2_H_2_. This study demonstrated that acetylene adsorption/desorption is also accomplished through molecular sieving. There is little research on MOF materials in the field of fruit and vegetable packaging for the adsorption/desorption of acetylene and ethephon, so the research focus can be placed on the fruit and vegetable ripening molecules in the future.

**Table 1 materials-16-03406-t001:** Metal–organic frameworks for food packaging application.

Metal–Organic Frameworks	Active Compound	Synthesis Method	Food	Outcomes	Reference
Cyclodextrin-based MOF	Hexanal	Vapor diffusion method	Mango	Shelf-life was extended to 15 days	Moussa et al. [54]
Single-walled nickel–organic framework	Hexanal	—	Banana	Banana placed in a 1-L MOF jar showed no sign of spoilage until day 30; in the control a dark spot was observed on day 9 of storage	Li et al. [55]
MIL-101@CMFP and UIO-66@CMFP	Curcumin	—	Pitaya	Curcumin-loaded nano-metal–organic framework extended the shelf-life of pitaya to 6 days; the control showed signs of spoilage on day 2 of storage	Huang et al. [56]
Electrospun pullulan/polyvinyl alcohol nanofibers incorporated in a porphyrin MOF	Thymol	—	Fresh grapes and strawberries	Grapes wrapped in the MOF showed no spoilage for 7 days; control showed rot on day 7 Strawberries remained fresh for 7 days when wrapped in the MOF; control showed mold growths	Min et al. [57]
Silver-based MOF	Chitosan	One-pot synthesis method	Pitaya	Spraying metal–organic framework solution on pitaya maintained its freshness for 14 days; control showed mold growth on day 7	Zhang et al. [58]
Copper terephthalate MOF	Chitosan	Solvothermal	Bananas and avocados	The shelf-life of bananas and avocados was extended	Zhang et al. [17]

**Table 2 materials-16-03406-t002:** Effect of ethylene and sensitivity for fresh fruits and vegetables.

Fresh Food Type	Ethylene Sensitivity	Effect of Ethylene
Apples	Very high (>100 mL kg^−1^h^−1^)	Brown, soften
Passion fruit	Very high (>100 mL kg^−1^ h^−1^)	Decay
Peaches	High (10–100 mL kg^−1^ h^−1^)	Soften
Pears	High (10–100 mL kg^−1^ h^−1^)	Fermentation, mildew
Banana	Medium (1–10 mL kg^−1^ h^−1^)	Decay
Mango	Medium (1–10 mL kg^−1^ h^−1^)	Spot
Cherries	Low (<1.0 mL kg^−1^ h^−1^)	Soften
Grapes	Low (<1.0 mL kg^−1^ h^−1^)	Decay
Broccoli	Low (<1.0 mL kg^−1^ h^−1^)	Mildew
Cauliflowers	Low (<1.0 mL kg^−1^ h^−1^)	Mildew

**Table 3 materials-16-03406-t003:** Overview of the effects of ethylene related to quality.

Ethylene Effect	Affected Organ	Fresh Food Type	Reference
Abscission	Bunch Stalk Calyx	Cherry Tomato Muskmelon	Beno-Moualem et al. [59] Dong et al. [60]
Sprouting	Tubercle, bulb	Persimmon Potato	Salvador et al. [61] Brasil and Siddiqui [4]
Color	Yellowing Stem browning	Onion Broccoli Sweet cherry	Bufler [62] Suzuki et al. [63] Zhao et al. [64]
Off-flavors	Volatiles	Banana	Cecchini et al. [65]
Toughness	Lignification	Asparagus	Toscano et al. [66]
Bitterness	Isocoumarin	Carrot, lettuce	Fan et al. [67]

## 4. Conclusions and Perspectives

MOFs have been proven to be an efficient component of food packaging due to the possibility of tailoring their structural space through the selection of the metal ions and the organic linker ligands, and their strong ethylene adsorption capacity. However, there are still big challenges in scaling up their use in large-scale food packaging application: the selectivity of separating similar gas molecules through MOFs is limited [68]; the fabrication of MOF-based packaging materials is difficult to integrate into large-scale industry methods; and metal ions and organic ligand functional groups are toxic, which requires more safety characterization for direct application in food packaging [69].

To solve these problems, the design of MOF-based packaging should be improved, and the mechanism of their action should be thoroughly investigated. For example, by using natural biomolecules as ligands and environmentally friendly transition metals or non-toxic metal ions, the toxicity of such packaging can be avoided and the materials can be made to be recyclable. Continuous advances in the field of microporous and mesoporous MOF materials can facilitate the development of food packaging that can more accurately differentiate between similar gas molecules. Further modification and design of such materials can help to improve performance; for example, the synthesis of MOFs can be achieved by choosing safer and less costly materials and production methods with lower energy consumption.

Overall, the use of MOF materials has strong potential for ensuring food safety and quality, and to extend the shelf-life of fruits and vegetables. In this review, the synthetic pathways for producing MOFs and the effects of ethylene on the ripening of fruits and vegetables have been outlined. Furthermore, the capability of MOF-based packaging to control ethylene adsorption/desorption has been discussed. The biocompatibility and non-reactivity of MOFs have increased the demand for these materials in the food packaging field. Although there are remaining drawbacks to this technology, such as precisely controlling the volume and pore size of MOFs following their modification for food packaging, MOFs and their composites still have promising applications as functional coatings for intelligent food packaging. Therefore, the development of multifunctional MOF materials with excellent sensing, stability, adsorption, and selectivity will assist in the formulation of advanced food packaging to aid the preservation of agriculture foods.

## Figures and Tables

**Figure 1 materials-16-03406-f001:**
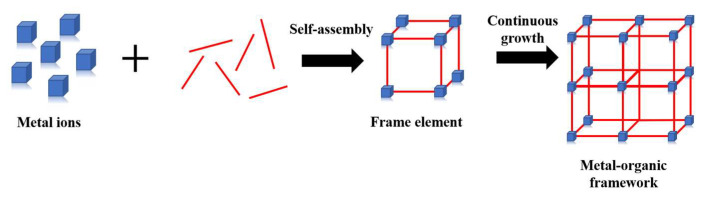
The process of MOF formation.

**Figure 2 materials-16-03406-f002:**
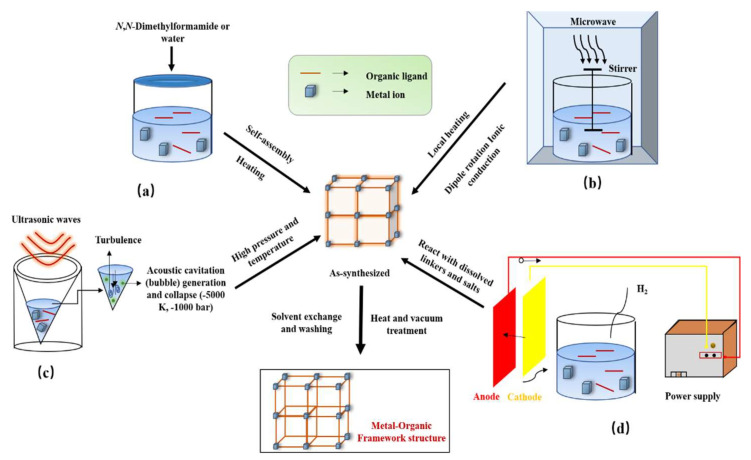
Common methods for synthesizing MOFs: (**a**) hydrothermal or solvothermal synthesis, (**b**) microwave-assisted synthesis, (**c**) sonochemical synthesis, (**d**) electrochemical synthesis.

## Data Availability

Not applicable.

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
