# Peer review of "MOF-Based Active Packaging Materials for Extending Post-Harvest Shelf-Life of Fruits and Vegetables"

_materials, 2023, doi:10.3390/ma16093406_

Round 1

Reviewer 1 Report

Title : MOF-based active packaging materials for extending post-harvest shelf-life of fruits  and vegetables

Authors :  Yabo Fu et al

Manuscript accounts the use of MOF-based materials in packaging of fresh fruits and vegetables and demand of extension of shelf-life assuring preservation of food quality and avoid food waste.  Microbial and pathogen growth are prevented or delaying of ethylene production thus ripening period is elongated. Authors were demanding that MOFs in package encapsulated ethylene, and enhanced the shelf-life of post-harvest fruits and vegetables. The ethylene adsorption/desorption mechanism of MOF-based packaging and its role in fruit and vegetable preservation is analysed.

Review is aimless. Title is very focused while the manuscript consumes sufficient space with peripheral gossiping those are neither analysed critically nor significantly useful to specialised readers. For example, fabrication of package that is very rudimentary and college level information; different synthetic process of MOFs (which is also incomplete !) and graduate level knowledge. Then, what is new and original for peer reviewed Research?

Although Ethylene is known as the “fruit-ripening hormone” but other chemicals acetylene, ethephon etc are also efficient fruit ripening molecules. So, only ethylene capture can’t be solution to make fruits, vegetables etc. fresh.

How long packaged MOFs restrain adsorbed Ethylene in confined space and what is ultimate fate? Will not ethylene slowly desorbed? How will indicate farmers the level of Ethylene adsorption, saturation and completion?

This manuscript is not suitable for publication as such in Materials.

Authors may focus to the selection of the molecules to prepare MOFs those may invite olefins (Ethylene, acetylene) to adsorb and develop towards design principle specially to olefins.

Authors may submit to “Coordination Polymers: Design Guidelines and Materials Perspective” to be published by Molecules, a special issue going to be published by MDPI.

Author Response

Response to reviewers

Dear reviewer,

Thank you very much for your comments and advice. These opinions help to improve academic rigor of our article. We have tried our best to improve the manuscript. Furthermore, we would like to show the details as follows:

Comment 1:

Review is aimless. Title is very focused while the manuscript consumes sufficient space with peripheral gossiping those are neither analyzed critically nor significantly useful to specialized readers.

Reply:

This article mainly focuses on the related application of MOF material packaging in extending the shelf life of fruits and vegetables. we focus on the active packaging of MOF materials. For scholars engaged in the packaging industry, the emergence of MOF materials brings them new options. At present, the fruit and vegetable packaging industry is really the most important thing to take measures, the integration of interdisciplinary disciplines will bring more choices, which is the focus of this article.

Comment 2:

For example, fabrication of package that is very rudimentary and college level information; different synthetic process of MOFs (which is also incomplete!) and graduate level knowledge. Then, what is new and original for peer reviewed research?

Reply:

Packaging is the necessary knowledge of production, but also the foundation. At the same time, for the synthesis of MOF materials, this paper does not list all the synthesis methods, but summarizes the topic of the paper. Due to the limitation of space, four synthesis methods that are widely studied at present are selected. This paper focuses on the adsorption/desorption mechanism of MOF materials on ethylene in active fruit and vegetable packaging and the embedding packaging method. This article has been focusing on active packaging. It is of certain significance to the scholars engaged in packaging research.

Comment 3:

Although Ethylene is known as the “fruit-ripening hormone” but other chemicals acetylene, ethephon etc are also efficient fruit ripening molecules. So, only ethylene capture can’t be solution to make fruits, vegetables etc. fresh.

Reply:

In fact, factors affecting the shelf life of fruits and vegetables are oxygen, microorganisms, and you mentioned ethephon, acetylene and other substances. These substances can affect the shelf life of fruits and vegetables. Ethylene is one of the most important influencing factors. Due to space constraints, other substances are not covered in too much space.

Comment 4:

How long packaged MOFs restrain adsorbed Ethylene in confined space and what is ultimate fate? Will not ethylene slowly desorbed?

Reply:

We would like to give an example of a reference (Zhang, W.; Banerjee, D.; Liu, J.; Schaef, H.T.; Crum, J.V.; Fernandez, C.A.; Kukkadapu, R.K.; Nie, Z.; Nune, S.K.; Motkuri, R.K.; et al. Redox-Active Metal-Organic Composites for Highly Selective Oxygen Separation Applications. Adv. Mater.2016, 28, 3572-+, doi:10.1002/adma.201600259.). The release rate was highest when the MOF-ethylene was first placed into a new container filled with air. The decrease in release rate over time could be attributed to the depletion of ethylene stored in the MOF-ethylene. Upon the loading of MOF with ethylene at ambient temperature and pressure according to the method described earlier, a total of 654 µL of ethylene was absorbed into the internal pores and external gaps in the 50 mg portion of MOF-ethylene. In the release kinetic study, 627 µL of ethylene was released into the container after 180 minutes, which accounted for 95.8% of the total absorbed ethylene in MOF-ethylene.

Comment 5:

How will indicate farmers the level of Ethylene adsorption, saturation and completion?

Reply:

The amount of MOF material is estimated according to the type of fruit and vegetable placed in the limited packaging space, according to the ethylene release and level, to guide the use. More systematic studies are needed in the future to quantify.

Comment 6:

This manuscript is not suitable for publication as such in Materials.

Authors may focus to the selection of the molecules to prepare MOFs those may invite olefins (Ethylene, acetylene) to adsorb and develop towards design principle specially to olefins.

Authors may submit to “Coordination Polymers: Design Guidelines and Materials Perspective” to be published by Molecules, a special issue going to be published by MDPI.

Reply: Thank you for your comment. The subject of this review is around the MOF-based active packaging materials for extending post-harvest shelf-life of fruits and vegetables. Therefore, we still hope to submit the special issue (Green & Intelligent Printing or Packaging Materials in Light Industry) in the journal of Materials.

In addition to the above problems, we also made other modifications:

  1. First of all, we ignored this important reference (Sultana, A.; Kathuria, A.; Gaikwad, K. K., Metal–organic frameworks for active food packaging. A review. Environmental Chemistry Letters 2022, 20 (2), 1479-1495) and added the arrow meaning in the figure and improved Figure 3. At the same time, I added the research to the introduction to support the key point that MOFs materials can be used in fruit and vegetable packaging.

Please see the revised content (1. Introduction)

The excellent controlled release and loading capacity of MOFs has been demonstrated, but few works on its ethylene adsorption/desorption in food packaging application has been reported [11]. Notably, MOF Technologies teamed up with Decco Worldwide Post-Harvest Holdings, developed the first MOF-based packaging for ethylene adsorption "Trupick" in 2016, which prevents fruit and vegetables ripening in storage. Trupick works by releasing 1-methylcyclepropene (1-MCP), a synthetic plant growth regulator that slows down ripening. It was thought that a MOF would be ideal for the storage and release of 1-MCP because of their porous structures.

Figure3. Incorporation of MOFs in packaging material to preserve food quality: MOF-impregnated packaging, sachets containing an MOF, coating with an MOF [26]

(in pdf)

  1. I read this e-book(https://pubs.acs.org/doi/book/10.1021/acs.infocus.7e4004), which gave me great help and inspiration. I added some content and further expanded Figure 2 to increase arrows of meaning in all sorts of directions.

Please see the revised content (2.1Fabrication of packaging materials)

Researchers have developed a wide variety of synthetic methods, especially in terms of reactors and temperatures. One feature all MOF synthesis methods have in common is crystal growth. An insoluble MOF can be formed from either the solid state, a melt, or a solution. In all cases, an initial nucleation occurs, producing a seed from which smaller or larger crystals may be grown. A key parameter here is the kinetics, which is usually con-trolled through concentration and temperature [11].

Figure2. Common methods for synthesizing MOFs: (a) hydrothermal or solvothermal synthesis, (b) microwave-assisted synthesis, (c) sonochemical synthesis, (d) electrochemical synthesis

 (in pdf)

  1. We realized that there is a lack of study to support my opinions. So I added related applications in 3.1 and revised some expressions.

Please see the revised content (3.1Application of MOF packaging materials in fruit and vegetable preservation)

Ref 31:Jiang, L.; Liu, F.; Wang, F.; Zhang, H.; Kang, M. Development and Characterization of Zein-Based Active Packaging Films Containing Catechin Loaded β-Cyclodextrin Metal-Organic Frameworks. Food Packaging and Shelf Life 2022, 31, 100810, doi:10.1016/j.fpsl.2022.100810.

Ref 32:Zhao, X.; Shi, T.-J.; Liu, Y.-Y.; Chen, L. Porphyrinic Metal–Organic Framework-Loaded Polycaprolactone Compo-site Films with a High Photodynamic Antibacterial Activity for the Preservation of Fresh-Cut Apples. ACS Applied Polymer Materials 2022, 5, doi:10.1021/acsapm.2c01667.

Ref 33:Wang, H.; Lashkari, E.; Lim, H.; Zheng, C.; Emge, T.J.; Gong, Q.; Yam, K.; Li, J. The Moisture-Triggered Controlled Release of a Natural Food Preservative from a Microporous Metal-Organic Framework. Chem. Commun. 2016, 52, 2129–2132, doi:10.1039/c5cc09634k.

  1. I added related applications in 3.1.1.(https://doi.org/10.1016/j.fpsl.2023.101034 )

Please see the revised content (3.1.1Application of MOFs in ethylene adsorption/desorption of fruit and vegetable packaging)

Recently,Zhang’s team evaluated MOF’s potential to be embedded in packaging films as ethylene adsorber, this study indicates that MgF-embedded LDPE packages could effectively delay banana ripening and extended their shelf life. Significantly, MgF has promising potential as an ethylene adsorber, justifying further work to investigate its application for fresh produce shelf-life extension, this study verified its feasibility under simulated in-transit condition.

In addition, there are some minor problems revised in the article. Thank you so much for your attention and time. Looking forward to your reply!

Dr. Yabo Fu

Yours sincerely,

28 Mar., 2023

Reviewer 2 Report

This review article has some nice features but unfortunately disregards that in 2016 fruit and vegetable storage was announced as the first commercial application of MOFs by MOF Technologies. This story needs to be incorporated. Also this key reference seems to be missing: Sultana, A.; Kathuria, A.; Gaikwad, K. K., Metal–organic frameworks for active food packaging. A review. Environmental Chemistry Letters 2022, 20 (2), 1479-1495.

Other issues:

Ref 10 seems to have nothing to do with MOFs?

Ref 11 and 12 are outdated an unhelpful for a general introduction. A recent ACS in Focus ebook gives a good introduction to all aspects of MOFs. https://pubs.acs.org/doi/book/10.1021/acs.infocus.7e4004

The excellent controlled release and loading capacity of MOFs has been demonstrated, but their application in the adsorption/desorption of ethylene in food packaging has only rarely been reported. 

Not sure about this. See my first point. 

An MOF is formed by the bonding of organic linkers and metal oxidation under the action of energy. ” This is not correct, please revise.

Section 2.1. Fabrication of packaging materials 

Is about MOF synthesis. Ref 16 is also outdated. This section should cite the ACS in Focus ebook.

2.1.1. Hydrothermal or solvothermal synthesis 

Misses the point of high pressure used.

Synthesis examples cited should be relevant for the subject of the article or omitted.

MOFs are highly ordered crystalline materials” They do not need to be crystalline. Please see the IUPAC recommendations.

 Chopra’s team [25] filled polyolefin bags 200 with Basolite C 300 loaded with,

Loaded with what?

Figure 3 has arrows in all sorts of directions, what do they mean? Please make a better figure. Also this figure comes very close to plagiarizing Fig. 6 of the non cited Sultanan article.

Recent studies have highlighted MOFs as potentially active agents in fruit and vegetable preservation because of their excellent oxygen scavenging, antibacterial activity, moisture absorption, and ethylene removal properties. ” Which studies?

Banerjee et 239 al. [31]developed MOFs that can be used to encapsulate ethylene and then release it in a controlled manner at the desired stage. ” The title of ref 31 is “Potential of Metal-Organic Frameworks for Separation of Xenon and Krypton.” Probably not the correct reference.

Ethylene is an invisible, colorless and odorless gaseous plant hormone that plays an 283 important role in the growth, development and storage of fruits and vegetables at ppm 284 or even ppb concentrations [36]. ” repetion

 organic metal framework ” please use correct terminology

Please revise:

Chui, null; Lo, null; Charmant, null; Orpen, null; Williams, null A Chemically Functionalizable Na- 435 noporous Material. Science 1999283, 1148–1150, doi:10.1126/science.283.5405.1148. 

Author Response

Response to reviewers

Dear reviewer,

Thank you very much for your comments and professional advice. These opinions help to improve academic rigor of our article. Based on your suggestion and request, we have made corrected modifications on the revised manuscript. We hope that our work can be improved again. Furthermore, we would like to show the details as follows:

Comment 1:

This review article has some nice features but unfortunately disregards that in 2016 fruit and vegetable storage was announced as the first commercial application of MOFs by MOF Technologies. This story needs to be incorporated. Also this key reference seems to be missing: Sultana, A.; Kathuria, A.; Gaikwad, K. K., Metal–organic frameworks for active food packaging. A review. Environmental Chemistry Letters 2022, 20 (2), 1479-1495.

Reply:

I ignored this important reference (Sultana, A.; Kathuria, A.; Gaikwad, K. K., Metal–organic frameworks for active food packaging. A review. Environmental Chemistry Letters 2022, 20 (2), 1479-1495) and added it to Figure 3. At the same time, we added the research you mentioned to the introduction to support the key point that MOFs materials can be used in fruit and vegetable packaging.

Please see the revised content (1. Introduction):

Notably, MOF Technologies teamed up with Decco Worldwide Post-Harvest Holdings, developed the first MOF-based packaging for ethylene adsorption "Trupick" in 2016, which prevents fruit and vegetables ripening in storage. Trupick works by releasing 1-methylcyclepropene (1-MCP), a synthetic plant growth regulator that slows down ripening. It was thought that a MOF would be ideal for the storage and release of 1-MCP because of their porous structures.

Comment 2:

Ref 10 seems to have nothing to do with MOFs?

Reply:

Thank you very much for your suggestions. This study focuses on the harvesting of antimicrobial peptides and its applications in food packaging. In this study, the author explained the application of such materials as Antimicrobial peptide(AMP) in active packaging, which can be realized in different ways. MOF materials can be selected from one or more of them, so I think Ref 10 should be added here.

Comment 3:

Ref 11 and 12 are outdated an unhelpful for a general introduction. A recent ACS in Focus ebook gives a good introduction to all aspects of MOFs. https://pubs.acs.org/doi/book/10.1021/acs.infocus.7e4004.

Reply:

The e-book does provide a detailed introduction to the MOF material, which we cite.

Please see the revised content (1. Introduction):

Since organic building blocks need to contain groups with accessible Free-electron pairs that can bind into metal ions and organize these ions in geometric shapes to produce networks in 2D or 3D rather than discrete molecular units [11].

Comment 4:

“The excellent controlled release and loading capacity of MOFs has been demonstrated, but their application in the adsorption/desorption of ethylene in food packaging has only rarely been reported. ”

Not sure about this. See my first point. 

Reply:

Please see the revised content (1. Introduction):

Original text:The excellent controlled release and loading capacity of MOFs has been demonstrated, but their application in the adsorption/desorption of ethylene in food packaging has only rarely been reported. 

Edited text: The excellent controlled release and loading capacity of MOFs has been demonstrated, but few works on its ethylene adsorption/desorption in food packaging application has been reported.

Comment 5:

An MOF is formed by the bonding of organic linkers and metal oxidation under the action of energy. ” This is not correct, please revise.

Reply:

Thank you very much for your suggestions. I revised a more combined with the content of the below.

Please see the revised content (2.1. Fabrication of packaging materials):

Original text: An MOF is formed by the bonding of organic linkers and metal oxidation under the action of energy.

Edited text: Researchers have developed a wide variety of synthetic methods, especially in terms of reactors and temperatures.

Comment 6:

Is about MOF synthesis. Ref 16 is also outdated. This section should cite the ACS in Focus ebook.

Reply:

Thank you very much for your suggestions. The content has been added in the article.

Please see the revised content (2.1. Fabrication of packaging materials):

One feature all MOF synthesis methods have in common is crystal growth. An insoluble MOF can be formed from either the solid state, a melt, or a solution. In all cases, an initial nucleation occurs, producing a seed from which smaller or larger crystals may be grown. A key parameter here is the kinetics, which is usually controlled through concentration and temperature [11].

Comment 7:

Misses the point of high pressure used.

Reply:

Thank you very much for your suggestions. I added this point in the article.

Please see the revised content (2.1.1. Hydrothermal or solvothermal synthesis):

Original text: The mixture is then transferred to a high-temperature reactor, cooled to room temperature at the end of the reaction.

Edited text: The mixture is then transferred to a high-temperature and high-pressure reactor [15], cooled to room temperature at the end of the reaction.

Comment 8:

Synthesis examples cited should be relevant for the subject of the article or omitted.

Reply:

The examples cited should be consistent with the subject of the article as far as possible. For example, the catalyst mentioned could be used in active packaging in the future, and the adsorption of bisphenol could be used in the detection food packaging materials.

Comment 9:

“MOFs are highly ordered crystalline materials” They do not need to be crystalline. Please see the IUPAC recommendations.

Reply:

Thank you very much for your suggestions. I revised the expression of this sentence.

Please see the revised content (2.2 Forms incorporated in food packaging):

Original text: MOFs are highly ordered crystalline materials and can be incorporated in food pack-aging in three ways.

Edited text: MOFs materials can be incorporated in food packaging in three ways.

Comment 10:

“ Chopra’s team [25] filled polyolefin bags 200 with Basolite C 300 loaded with,”

Loaded with what?

Reply:

This is a minor error in writing and has been corrected.

Please see the revised content (2.2.2 Sachets):

Original text: Chopra’s team [25] filled polyolefin bags 200 with Basolite C 300 loaded with,

Edited text: Chopra’s team[24]filled polyolefin bags with Basolite C 300 and added the bags to banana packaging.

Comment 11:

Figure 3 has arrows in all sorts of directions, what do they mean? Please make a better figure. Also this figure comes very close to plagiarizing Fig. 6 of the non cited Sultanan article.

Reply:

<1>Thank you very much for your suggestions. Based on your suggestions, I added the arrow meaning in the figure and improved Figure 3. And I consulted Sultanan's article and on the basis of the modified. I should have highlighted it in the article.

Figure 3 (in pdf)

<2>Besides, I further improved in figure 2 and added the meaning of the arrow in the figure 2.

Figure 2 (in pdf)

Comment 12:

“Recent studies have highlighted MOFs as potentially active agents in fruit and vegetable preservation because of their excellent oxygen scavenging, antibacterial activity, moisture absorption, and ethylene removal properties. ” Which studies?

Reply:

I realized that there is a lack of study to support my opinions. So I added related references in 3.1.

Please see the revised content (3.1. Application of MOF packaging materials in fruit and vegetable preservation):

Ref 31:Jiang, L.; Liu, F.; Wang, F.; Zhang, H.; Kang, M. Development and Characterization of Zein-Based Active Packaging Films Containing Catechin Loaded β-Cyclodextrin Metal-Organic Frameworks. Food Packaging and Shelf Life 2022, 31, 100810, doi:10.1016/j.fpsl.2022.100810.

Ref 32:Zhao, X.; Shi, T.-J.; Liu, Y.-Y.; Chen, L. Porphyrinic Metal–Organic Framework-Loaded Polycaprolactone Compo-site Films with a High Photodynamic Antibacterial Activity for the Preservation of Fresh-Cut Apples. ACS Applied Polymer Materials 2022, 5, doi:10.1021/acsapm.2c01667.

Ref 33:Wang, H.; Lashkari, E.; Lim, H.; Zheng, C.; Emge, T.J.; Gong, Q.; Yam, K.; Li, J. The Moisture-Triggered Controlled Release of a Natural Food Preservative from a Microporous Metal-Organic Framework. Chem. Commun. 2016, 52, 2129–2132, doi:10.1039/c5cc09634k.

Comment 13:

“Banerjee et 239 al. [31] developed MOFs that can be used to encapsulate ethylene and then release it in a controlled manner at the desired stage. ” The title of ref 31 is “Potential of Metal-Organic Frameworks for Separation of Xenon and Krypton.” Probably not the correct reference.

Reply:

Thank you very much for your suggestions. I noticed that the wrong reference was used in the article and I have revised it in the article.

Please see the revised content (3.1.1. Application of MOFs in ethylene adsorption/desorption of fruit and vegetable packaging):

Original text: Banerjee et 239 al. [31] developed MOFs that can be used to encapsulate ethylene and then release it in a controlled manner at the desired stage.

Edited text: Zhang et al. [34] developed MOFs that can be used to encapsulate ethylene and then release it in a controlled manner at the desired stage.

Comment 14:

“Ethylene is an invisible, colorless and odorless gaseous plant hormone that plays an 283 important role in the growth, development and storage of fruits and vegetables at ppm 284 or even ppb concentrations [36]. ”repletion.

Reply:

I think this sentence overlaps with the previous introduction and it’s not very relevant to this paragraph, so I decided to delete this sentence.

Comment 15:

“organic metal framework ”please use correct terminology.

Reply:

 Thank you very much for your suggestions. We 've corrected the typo and apologize for our error

Please see the revised content (3.2.1. Application of MOFs in ethylene adsorption/desorption of fruit and vegetable packaging):

Original text: organic metal framework

Edited text: metal-organic framework

Comment 16:

Please revise:

Chui, null; Lo, null; Charmant, null; Orpen, null; Williams, null A Chemically Functionalizable Na- 435 noporous Material. Science 1999283, 1148–1150,

doi:10.1126/science.283.5405.1148.

Reply:

Thank you very much for your suggestions. I have deleted this reference.

In addition, there are some minor problems revised in the article. Thank you so much for your attention and time. Looking forward to your reply!

Dr. Yabo Fu

Yours sincerely,

28 Mar., 2023

Reviewer 3 Report

This manuscript gives a nice overview on MOF-based packaging materials for fruits and vegetables. It is well-suited for any newcomer in this field, however, without going too detailed into the fundamental understanding of microscopic mechanisms. I suggest to include in the discussion the results of a very recent paper, https://doi.org/10.1016/j.fpsl.2023.101034 which identified under several MOFs a specific MOF best suited for ethylene adsorption.

Author Response

Response to reviewers

Dear reviewer,

Thank you very much for your comments and professional advice. These opinions help to improve academic rigor of our article. Based on your suggestion and request, we have made corrected modifications on the revised manuscript. We hope that our work can be improved again. Furthermore, we would like to show the details as follows:

Comment:

This manuscript gives a nice overview on MOF-based packaging materials for fruits and vegetables. It is well-suited for any newcomer in this field, however, without going too detailed into the fundamental understanding of microscopic mechanisms. I suggest to include in the discussion the results of a very recent paper, https://doi.org/10.1016/j.fpsl.2023.101034 which identified under several MOFs a specific MOF best suited for ethylene adsorption.

Reply:

<1>Thank you very much for your suggestions. I added related applications in 3.1.1.

Please see the revised content (3.1.1. Application of MOFs in ethylene adsorption/desorption of fruit and vegetable packaging):

Recently, Zhang’s team evaluated MOF’s potential to be embedded in packaging films as ethylene adsorber, this study indicates that MgF-embedded LDPE packages could effectively delay banana ripening and extended their shelf life. Significantly, MgF has promising potential as an ethylene adsorber, justifying further work to investigate its application for fresh produce shelf life extension, this study verified its feasibility under simulated in-transit condition.

<2>Besides, according to your suggestions, I have read relevant review and expanded the contents of 3.1.

Please see the revised content (3.1. Application of MOF packaging materials in fruit and vegetable preservation)

Ref 31:Jiang, L.; Liu, F.; Wang, F.; Zhang, H.; Kang, M. Development and Characterization of Zein-Based Active Packaging Films Containing Catechin Loaded β-Cyclodextrin Metal-Organic Frameworks. Food Packaging and Shelf Life 2022, 31, 100810, doi:10.1016/j.fpsl.2022.100810.

Ref 32:Zhao, X.; Shi, T.-J.; Liu, Y.-Y.; Chen, L. Porphyrinic Metal–Organic Framework-Loaded Polycaprolactone Compo-site Films with a High Photodynamic Antibacterial Activity for the Preservation of Fresh-Cut Apples. ACS Applied Polymer Materials 2022, 5, doi:10.1021/acsapm.2c01667.

Ref 33:Wang, H.; Lashkari, E.; Lim, H.; Zheng, C.; Emge, T.J.; Gong, Q.; Yam, K.; Li, J. The Moisture-Triggered Controlled Release of a Natural Food Preservative from a Microporous Metal-Organic Framework. Chem. Commun. 2016, 52, 2129–2132, doi:10.1039/c5cc09634k.

In addition, we also made the following modifications:

  1. we added the research you mentioned to the introduction to support the key point that MOFs materials can be used in fruit and vegetable packaging.

Please see the revised content (1. Introduction):

Notably, MOF Technologies teamed up with Decco Worldwide Post-Harvest Holdings, developed the first MOF-based packaging for ethylene adsorption "Trupick" in 2016, which prevents fruit and vegetables ripening in storage. Trupick works by releasing 1-methylcyclepropene (1-MCP), a synthetic plant growth regulator that slows down ripening.  It was thought that a MOF would be ideal for the storage and release of 1-MCP because of their porous structures.

  1. I read this e-book(https://pubs.acs.org/doi/book/10.1021/acs.infocus.7e4004., which gave me great help and inspiration. I improved the relevant content here.

Please see the revised content (1. Introduction):

Since organic building blocks need to contain groups with accessible Free-electron pairs that can bind into metal ions and organize these ions in geometric shapes to produce networks in 2D or 3D rather than discrete molecular units [11].

Please see the revised content (2.1. Fabrication of packaging materials):

One feature all MOF synthesis methods have in common is crystal growth. An insoluble MOF can be formed from either the solid state, a melt, or a solution. In all cases, an initial nucleation occurs, producing a seed from which smaller or larger crystals may be grown. A key parameter here is the kinetics, which is usually controlled through concentration and temperature [11].

  1. I added the arrow meaning in the figure and improved Figure 3. And I consulted Sultanan's article and on the basis of the modified. I should have highlighted it in the article. Besides, I further improved in figure 2 and added the meaning of the arrow in the figure 2.

Figure 3 : (in pdf )

Figure 2 : (in pdf)

In addition, there are some minor problems highlighted in the article. Thank you so much for your attention and time. Looking forward to your reply!

Dr. Yabo Fu

Yours sincerely,

28 Mar., 2023

Round 2

Reviewer 1 Report

Authors attempted to revise the manuscripts with jugglery of words but not in spirit. No new information or knowledge will support to the readers. So, I am not recommending its publication.   

Author Response

Dear professor,

Thanks for your comments, we are highly appreciated. As for the novelty, MOF materials have attracted extensive attention from researchers. From the introduction of the structure of MOF materials, the causes and mechanisms of synthesis and multifunctional applications, as well as the published reviews of energy storage properties, catalytic properties, separation properties, biological properties, hydrogen storage properties, optical properties and magnetic properties, MOF materials have been well known to researchers. However, there are few studies on MOF materials in  packaging industry. Although there are some reports focusing on packaging, the focus on food packaging has been rarely explored. In this paper, we summarized the application of MOF material in fruit and vegetable packaging, discussed the adsorption and desorption of ethylene, and prospected the development of MOF material in the future packaging field. Thanks again to the reviewer on suggesting to further improve this manuscript, we have studied comments carefully and have made corresponding corrections which we hope meet with approval.

The purpose of this review is to study the experimental methods and material systems of different researchers, including the application mechanism of MOF materials in ethylene adsorption/desorption, and lay a solid foundation for our future research papers. In this paper, the physical and chemical properties of MOF are not studied in detail. Due to the limitation of space, we focus more on the application in the field of packaging, and mainly summarize the achievements made by researchers in the field of food packaging in recent years. Through reading literature, it is found that MOF materials are applied more in the industrial field, but less in the packaging light industry, and the cross integration of different disciplines can help researchers solve more problems.

According to your suggestion, we have consulted the latest literature and added the content to the article (3.2.1), The details are as follows: ‘Although ethylene is known as the fruit-ripening hormone, other chemicals such as acetylene are also effective fruit ripening molecules. Yang et al.[53] obtained materials boost molecular sieving-based separation of CO2/C2H2 and realize the over-whelming adsorption of CO2 over C2H2. This study demonstrated that acetylene adsorp-tion/desorption is also accomplished through molecular sieve. There are few researches on MOF materials in the field of fruit and vegetable packaging for the adsorp-tion/desorption of acetylene and ethephon, so the research focus can be placed on the fruit and vegetable ripening molecules in the future.’

MOFs have been proven to be an efficient component of food packaging due to the possibility of tailoring their structural space through the selection of the metal ions and the organic linker ligands, and their strong adsorption capacity. We selected MOF material as the substrate for the following research on fruit and vegetable packaging. Since its potential has been proved by published reviews, we were prompted to carry out further research, including the adsorption properties of related ripening molecules such as ethylene and acetylene.
